# p53 Forms Redox-Dependent Protein–Protein Interactions through Cysteine 277

**DOI:** 10.3390/antiox10101578

**Published:** 2021-10-06

**Authors:** Tao Shi, Paulien E. Polderman, Marc Pagès-Gallego, Robert M. van Es, Harmjan R. Vos, Boudewijn M. T. Burgering, Tobias B. Dansen

**Affiliations:** 1Center for Molecular Medicine, Molecular Cancer Research, University Medical Center Utrecht, Universiteitsweg 100, 3584 CG Utrecht, The Netherlands; t.shi@umcutrecht.nl (T.S.); p.e.polderman@umcutrecht.nl (P.E.P.); m.pagesgallego@umcutrecht.nl (M.P.-G.); r.m.vanes-4@umcutrecht.nl (R.M.v.E.); h.r.vos-3@umcutrecht.nl (H.R.V.); b.m.t.burgering@umcutrecht.nl (B.M.T.B.); 2Oncode Institute, University Medical Center Utrecht, Universiteitsweg 100, 3584 CG Utrecht, The Netherlands

**Keywords:** p53, cysteine oxidation, disulfide, protein–protein interaction

## Abstract

Reversible cysteine oxidation plays an essential role in redox signaling by reversibly altering protein structure and function. Cysteine oxidation may lead to intra- and intermolecular disulfide formation, and the latter can drastically stabilize protein–protein interactions in a more oxidizing milieu. The activity of the tumor suppressor p53 is regulated at multiple levels, including various post-translational modification (PTM) and protein–protein interactions. In the past few decades, p53 has been shown to be a redox-sensitive protein, and undergoes reversible cysteine oxidation both in vitro and in vivo. It is not clear, however, whether p53 also forms intermolecular disulfides with interacting proteins and whether these redox-dependent interactions contribute to the regulation of p53. In the present study, by combining (co-)immunoprecipitation, quantitative mass spectrometry and Western blot we found that p53 forms disulfide-dependent interactions with several proteins under oxidizing conditions. Cysteine 277 is required for most of the disulfide-dependent interactions of p53, including those with 14-3-3θ and 53BP1. These interaction partners may play a role in fine-tuning p53 activity under oxidizing conditions.

## 1. Introduction

The transcription factor p53 is a key player in the cellular stresses response and is activated by DNA damage, and by oncogenic, oxidative and metabolic stress [1]. p53 activation triggers cell cycle arrest and apoptosis, but is also involved in cellular survival programs through the induction of DNA damage repair and metabolic regulation. Collectively, these programs contribute to the maintenance of genome integrity and protect the organism from over-proliferation of cells that carry oncogenic mutations. p53 stabilization and function are controlled by post-translational modifications (PTMs) such as phosphorylation, acetylation, ubiquitination and methylation that may vary depending on the type of stress [2].

Reactive oxygen species (ROS), mainly in the form of hydrogen peroxide (H_2_O_2_), act as a second messenger in so-called redox signaling, which involves oxidative modification of cysteine thiol side chains to regulate the function of target proteins [3,4]. Oxidation of thiols to form sulfenic acid (S-OH) or disulfide (S-S-) is reversible by the cellular antioxidant system, enabling to switch the redox signal on and off. The extent of cysteine oxidation in the cellular proteome therefore depends on the rates of production and clearance of ROS and the rates of oxidation and reduction in thiols. A particular attractive mode of redox modification is the formation of intermolecular disulfides, because it can stabilize otherwise weak protein–protein interactions. In this way, protein function can be modified to an extent which correlates with the local redox environment. Intermolecular disulfide formation has been shown to play roles in signaling in species from yeast to human [5]. 

p53 has also been found to be oxidized on multiple cysteines upon oxidant treatment, both in vitro and in live cells. In vitro, C182 and C277 were identified to be reactive to the alkylating agent N-ethylmaleimide [6]. C182 can also form an intramolecular disulfide bond with one of the three Zinc-binding cysteines (C176, 238 and 242), resulting in the loss of Zinc and protein unfolding [7]. Consistently, Held et al. quantified the extent of site-specific reversible cysteine oxidation in endogenous p53 and found that both C182 and C277 were sensitive to the thiol oxidant diamide [8], but the exact type of reversible cysteine oxidation remained unknown. Here, we set out to study whether p53 forms disulfide-dependent intermolecular interactions upon oxidation. To this end we combined immunoprecipitation and quantitative Mass spectrometry on wild type and p53 cysteine mutants expressed in HEK293T cells to identify redox-dependent interaction partners of p53. Intriguingly, in line with the observations by Held et al., diamide but not H_2_O_2_ induced oxidative stress, which stimulated the formation of disulfide-dependent complexes with p53. Some well-known p53 regulators were among the identified disulfide-dependent binders, including 14-3-3θ and 53BP1, and these depended on the presence of C277. Nevertheless, the p53 277S mutant was still activated by Nutlin-3 treatment, oxidative signaling and DNA damage, suggesting that the identified disulfide-dependent interactors are not critical for p53 function per se. We propose that the observed covalent interactions with p53 could be involved in fine tuning the spatiotemporal p53 response by stabilization otherwise weak protein–protein interactions under oxidizing conditions.

## 2. Materials and Methods

### 2.1. Constructs, Reagents and Antibodies

The pDONR223-p53 WT plasmid was a gift from Jesse Boehm, William Hahn and David Root (Addgene plasmid # 81754 [9]). pDON223-p53 Cysteine mutants (Cys to Ser or Ala) were generated by site-directed mutagenesis PCR using pDONR223-p53 WT as the template. The primers used for mutagenesis PCR are shown in Appendix A. N-terminally tagged Flag- and HA-p53 expression as well as doxycycline-inducible Flag-p53 WT and -C277S constructs were obtained by a Gateway cloning with pcDNA3 or pInducer20 backbones (pInducer20 was a gift from Stephen Elledge (Addgene plasmid # 44012) [10]). 

Diamide(D3648), hydrogen peroxide solution 30% (7722-84-1), Neocarzinostatin (NCS)(N9162), Auranofin (AFN) (A6733) and N-Ethylmaleimide (NEM)(E3876) were from Sigma. Nutlin-3(10004372) was from Sanbio. The 14-3-3θ siRNA(sc-29586) was from Santa Cruz Biotechnology (Dallas, TX, USA). 

Anti-Flag^®^M2 affinity gel (A220), anti-HA agarose (A2095), Anti-FLAG^®^(rabbit)(F7425) and anti-FlagM2 antibody (F1804) were from Sigma-Aldrich (Burlington, MA, USA). Antibodies against p53(DO-1), p21(M-19), 53BP1(H-300) and 14-3-3θ (5J20) were from Santa Cruz Technology (Dallas, TX, USA). Anti-pCHK2(Thr68) (CS2661) antibody was from Cell Signaling Technology (Danvers, MA, USA). Anti-GAPDH (MAB374) antibody was from EMD Millipore (Burlington, MA, USA). Anti-HA (12CA5) antibody was prepared in-house from hybridoma cell lines. Goat anti-mouse IgG-HPR (170-6516) and Goat anti-rabbit IgG-HRP (170-6515) were form Bio-Rad (Hercules, CA, USA). Fluorescence-conjugated secondary antibodies: IRDye 680RD goat anti-mouse IgG (925-68070), IRDye 800CW goat anti-mouse IgG (926-32210), IRDye 680 goat anti-rabbit IgG (926-32221) and IRDye 800CW goat anti-rabbit IgG (926-32211) from Li-Cor (Lincoln, NE, USA). 

### 2.2. Cell Culture

HEK293T, non-small cell lung cancer cells (NCI-H1299) (p53-deficient) cells were cultured in DMEM high-glucose (4,5g/L) containing 10% FBS, 2mM L-glutamine and 100 Units Penicillin-Streptomycin (All from Sigma Aldrich, Burlington, MA, USA), under a 6% CO_2_ atmosphere and at 37 °C. Transient transfections were performed using the polyethyleneimine (PEI) transfection reagent (Sigma Aldrich, Burlington, MA, USA). p53 KO RPE^Tert^ cells were a gift from René Medema [11], and cultured in DMEM/F-12 high-glucose supplemented with 10 % FBS and 100 U Penicillin-Streptomycin (Sigma Aldrich, Burlington, MA, USA) under a 6% CO_2_ atmosphere and at 37 °C. Doxycycline-inducible expression of Flag-p53 WT and C277S cells was generated by transduction with lentiviral constructs pInducer20-Flag-p53 WT and C277S in the p53-KO RPE^Tert^ cells, followed by the selection by Neomycin (400 µg/mL) for two weeks. The dox-inducible expression of Flag-p53 was confirmed by Western blot detection and polyclonal cells were used for subsequent experiments. 

### 2.3. Cell Lysis, Immunoprecipitation and Western Blot

For total lysates, cells seeded in 6-well dishes were directly scraped in loading sample buffer (Tris-HCI pH 6.8, 2% SDS, 5% 2-mercaptoethanol, 10% glycerol, 0.002% bromophenol blue). For immunoprecipitation experiments, HEK293T or H1299 cells were seeded in 10 cm dishes and transiently transfected with the indicated constructs. 48 h after transfection, cells were treated with diamide or H_2_O_2_ for the indicated time, followed by incubation with 100 mM *N*-ethylmaleimide (NEM) in PBS at 37 °C for 5 min to prevent post-lysis oxidation and to inactivate disulfide-reducing enzymes. Cells were scraped in the same NEM buffer and collected by centrifugation at 1500 rpm for 5 min. Cell pellets were resuspended in 1mL of lysis buffer containing 50 mM Tris pH 7.5, 1% Triton, 1.5 mM MgCl_2_, 1 mM EDTA, 100 mM NaCl supplemented with Aprotinin, Leupeptin, NaF and 100 mM Iodoacetamide to further prevent post-lysis oxidation.

Cell lysates were subsequently centrifuged at 14,000 rpm for 10 min. 50 µL of supernatants were taken as a control (‘input’) and the rest was used for immunoprecipitation and incubated with 15 µL of anti-FlagM2 or HA Affinity beads. After 2 h of incubation at 4 °C, beads were washed with wash buffer (50 mM Tris pH 7.5, 1% Triton, 1.5 mM MgCl_2_, 1 mM EDTA, 1 M NaCl supplemented with Aprotinin, Leupeptin and NaF) three times to minimize non-specific binding and enrich for disulfide-dependent interactions. After washing, samples were firstly resuspended in 1× non-reducing sample buffer (without β mercaptoethanol) and boiled at 95 °C for 10 min. Half of the samples were loaded for non-reducing sample detection and the rest half was added 5× reducing buffer (with β mercaptoethanol), boiled again at 95 °C for 5 min and loaded for reducing sample detection. 

For SDS-PAGE followed by Western blot, samples were run on 7.5 % or 10 % SDS-PAGE gels depending on the molecular weight of the proteins of interest. After that, proteins were transferred to a PVDF (polyvinylidene difluoride), nitrocellulose or immobilon-FL membrane (Millipore, Burlington, MA, USA) through a traditional wet transfer method. Membranes were blocked with 2% BSA in TBST for 1 h at 4 °C and then incubated with primary antibodies overnight at 4 °C, followed by washing with TBST solution before secondary antibody staining. Secondary antibody staining was performed using HRP or fluorescence-conjugated secondary antibodies for 1h at 4 °C. For imaging, membranes were washed again with TBST and subsequently analyzed on a FujiFilm LAS-3000 Luminescent Image Analyzer (for HRP) (Tokyo, Japan) or Amersham™ Typhoon™ Biomolecular Imager (GE Healthcare, Chicago, IL, USA) for fluorescence.

### 2.4. Sample Preparation for Mass Spectrometry

HEK293T cells were seeded in 15 cm dishes (4 replicates per condition) and transfected with 20 µg of Flag-p53, Flag-C182S,C277S or Flag-C182S DNA constructs. After 48 h, cells were treated with diamide for 15 min, followed by incubation with 100 mM NEM in PBS at 37 °C for 5 min to alkylate free thiols and prevent post-lysis oxidation. Cells were scraped in NEM buffer, and all replicates were collected in the same 15 mL tube followed by centrifugation at 1500 rpm for 5 min. Cell pellets were lysed in 8 mL of lysis buffer as describe in Section 2.3 and 1% of supernatant was taken as input. 80 µL of FlagM2 agarose beads were taken for immunoprecipitation against Flag following the procedure as described above in Section 2.3. After final cleaning of the beads, 1% beads solution was taken for Western blotting detection pre-MS and the rest was used for MS experiment. Proteins on beads were incubated with reduction and alkylation buffer (1 M Ammonium bicarbonate, 50 mM Acetonitrile, 10 mM TCEP, 40 mM CAA and 8M urea) at room temperature for 30 min, and then digested with 250 ng of trypsin overnight at 37 °C on a shaker. Peptides were then loaded on C18 StageTips and washed twice with 0.1% formic acid solution (diluted in water). Peptides on C18 StageTips are stable and can be stored at 4 °C up to one month. 

### 2.5. Mass Spectrometry

Mass spectrometry was performed as previously described [12]. Briefly, peptides were separated on a 30 cm pico-tip column (75 µm ID, New Objective) and were packed in-house with 3 µm aquapur gold C-18 material (Dr. Maisch) using a 140 min gradient (7–80%ACN, 0.1% FA), delivered by an easy-nLC 1000 (LC 120, Waltham, MA, USA, Thermo Scientific) and electro-sprayed directly into an Orbitrap Fusion Tribrid Mass Spectrometer (LC 120, Waltham, MA, USA, Thermo Scientific). This was then run in data-dependent mode with the resolution of the full scan set at 240,000, after which the top N peaks were selected for HCD fragmentation (30% collision energy) using the top speed option with a cycle time of 1 second with a target intensity of 1E4. The mass spectrometry proteomics data were submitted to ProteomeXchange via the PRIDE database with identifier PXD026893 [13].

### 2.6. Mass Spectrometry Data Analysis

The raw mass spectrometry files were processed using Maxquant software (version 1.5.2.8). The human protein database of UniProt was searched with both proteins and peptides (false discovery rate set to 1%). Data analysis regarding the identified proteins was further analyzed in R (version 3.6.1). Proteins were filtered for reverse hits and standard contaminants. Proteins for which less than 2 peptides were identified were also removed. Label-Free-Quantification (LFQ) values were log_2_-transformed, and the *proDA* (inference of protein differential abundance by probabilistic dropout analysis) model was used to impute missing values following data analysis [14]. Significant hits between conditions (e.g., CTRL vs. Diamide) were judged by at least 2-fold change in protein abundance with an adjusted *p*-value (Benjamini-Hochberg) smaller than 0.01. The *ggplot2* package was used to plot the data. The R scripts, raw and processed data are deposited in https://github.com/Taoshi2021/p53-oxidation (deposited on 17 June 2021).

### 2.7. Immunofluorescence Microscopy

RPE^Tert^ p53 KO cells expressing Doxycycline-inducible Flag-p53 WT and C277S were grown on glass coverslips in 6-well dishes and treated with Dox for 48 h. The cells were then fixed with 3.7% Formaldehyde solution at room temperature for 15 min, followed by permeabilization using 0.1% Triton for 5 min and subsequent blocking with 2% BSA (*w/v*) and purified goat IgG in 1:10,000 in PBS for 45 min at room temperature. The cells were then incubated with the primary antibody DO-1 against p53 at a final 1:500 dilution overnight, followed by 1 h incubation with a secondary antibody conjugated with Alexa fluor 488 (ThermoFisher Scientific, Waltham, MA, USA) and Hoechst 33,342 (Life Technologies, Carlsbad, CA, USA) after washing twice with PBS. All antibody staining was performed at 4 °C and in the dark. Finally, the coverslips were mounted in a drop of mounting medium and saved at 4 °C in the dark for further analysis. Imaging was performed on a Zeiss confocal microscope LSM880 and images were processed in Fiji (ImageJ) software.

### 2.8. Ubiquitination Assay 

HEK293T cells in 10 cm dishes were transiently transfected with the in the text indicated DNA constructs. After 48 h, cells were treated with H_2_O_2_ or diamide for 15 min, and then scraped in lysis buffer (100 mM NaH_2_PO_4_/Na_2_HPO_4_, 10 mM Tris, 8 M Urea, 10 mM NEM, 10 mM Imidazole and 0.2% Triton X-100, pH 8.0). Cell lysates were sonicated and then centrifuged at 10,000 rpm for 10 min. 50 µL of supernatant were taken as a control for input and the remainder was subjected to pulldown using Ni-NTA beads to enrich for His-Ubiquitin-tagged proteins. After 2 h of incubation at room temperature, beads were washed twice with wash buffer (100 mM NaH_2_PO_4_/Na_2_HPO_4_, 10 mM Tris, 8 M Urea, 10 mM Imidazole and 0.2% Triton X-100, pH 6.3), followed by one-time wash with elution buffer (100 mM NaCl, 20% glycerol, 20 mM Tris, 1 mM DTT and 10 mM Imidazole, pH 8.0). Ultimately, samples were resuspended in 1x reducing sample buffer for subsequent analysis.

### 2.9. RNA Isolation and qPCR

p53 target gene expression was analyzed by qPCR on RNA extracted from Dox-inducible expressing p53 WT and C277S in p53 KO RPE^Tert^ cells. Total RNA was isolated using a RNeasy kit (QIAGEN, Hilden, Germany). 500 ng of RNA was used for cDNA synthesis according to the manufacturer’s instructions using the iScript cDNA Synthesis Kit (Bio-Rad, Hercules, CA, USA). qPCR was performed with SYBR Green FastStart Master Mix in the CFX Connect Real-time PCR detection system (Bio-Rad, Hercules, CA, USA). The procedures were as follows: pre-denaturing at 95 °C for 10 min, followed by denaturing at 95 °C for 10 seconds, annealing at 58 °C for 10 s and extending at 72 °C for 30 s for 39 cycles. All the primers used for the qPCRs are shown in Appendix A.

### 2.10. Sequence Alignment 

The p53 protein sequence of vertebrate species and its paralogs (p63 and p73) were downloaded from ENSEMBL database [15]. Sequence alignment was performed in Jalview (version 15.0) software and was colored by the extent of conservation (threshold 15) [16]. Cysteines were subsequently colored in orange regardless of conservation status using Adobe Illustrator. 

### 2.11. Gene Ontology Enrichment Analysis

Gene Ontology (GO) enrichment analysis was performed using the online PANTHER Classification System (http://pantherdb.org/) (accessed on 17 April 2020). 162 proteins that were identified to significantly bind to wild-type p53 upon diamide treatment were selected for GO analysis with the annotation sets of ‘biological process’, ‘molecular function’ and ‘cellular component’. All genes (*Homo sapiens*) in the database were used as the reference list. *p* value was evaluated by the classic Fisher test, and the value lower 0.001 was a cutoff of significance.

## 3. Results

### 3.1. p53 Forms Intermolecular Disulfide-Dependent Complexes upon Oxidation

p53 has long been known to be prone to reversible oxidation on cysteines both in vitro and in vivo [8,17,18] (also reviewed in [19]), but the chemical identity of the reversible oxidation is generally lost during sample preparation. Reversible cysteine oxidation can result in intermolecular disulfide formation and the latter has been shown to play important roles in tuning protein function and signal transduction [5]. Intermolecular disulfide-dependent complexes can be detected based on migrational behavior with large mass-shifts on SDS-PAGE under non-reducing conditions followed by Western blot. Reduction in the same sample by beta-mercaptoethanol dissociates the complex and p53 migrates at its monomeric mass. Flag-p53 wildtype (WT) expressed in HEK293T cells indeed formed multiple redox-sensitive protein complexes upon treatment with the thiol-specific oxidant diamide (Figure 1A). Complex formation was dose-dependent and dissolved over time by the cellular antioxidant system (Figure 1A,B and Appendix A). A number of distinct bands was observed, suggesting that p53 forms complexes with a specific set of proteins rather than crosslink with proteins randomly. The extent of p53-containing disulfide-dependent protein complexes peaked 15 min after diamide treatment and was largely resolved 1 h after of diamide-addition, regardless of whether diamide was washed out or not (Figure 1B and Appendix A). Surprisingly, H_2_O_2_ did not induce clear p53 redox-dependent complexes even at concentrations well above those that induce a DNA damage response or upon prolonged treatment (Figure 1A and Appendix A). Although this is consistent with the previous finding that endogenous p53 is more susceptible to diamide-dependent oxidation as compared with H_2_O_2_-dependent oxidation [8], we looked further into this. Incubation of cells with H_2_O_2_ in PBS instead of complete media did not result in p53 S-S-dependent complex formation, ruling out that rapid clearance of H_2_O_2_ by media components prevents p53 oxidation (Appendix A). Continuous extracellular H_2_O_2_ production by addition of glucose oxidase to the culture media did result in p53 oxidation, albeit with lesser intensity than upon treatment with diamide (Appendix A). Interestingly, when HEK293T cells were cultured in low glucose rather than high glucose media, H_2_O_2_ did induce p53 redox-dependent complexes to a similar extent as diamide (Appendix A).

Diamide is a thiol-specific oxidizing agent that most probably induces redox signaling by lowering of the cellular reductive capacity, for instance through oxidation of glutathione [20]. We tested whether inhibition of the cellular thioredoxin (Trx) system, using the thioredoxin reductase (TrxR) inhibitor auranofin (AFN) also induced p53 redox-dependent complexes [21,22]. Indeed, AFN treatment also led to redox-dependent complex formation of p53 (Figure 1C). However, diamide and AFN did not result in identical patterns of disulfide-dependent interactions with p53 as judged by the protein shifts on the blot. There could be several molecular mechanisms that underlie these differences. 

### 3.2. p53 Forms Disulfide-Dependent Protein Complexes through C277

Next, we questioned which cysteine(s) in p53 is (are) involved in the observed intermolecular disulfide-dependent complexes. The p53 protein has ten cysteines that are all located in the DNA-binding domain (Figure 2A). Three of these (C176, C238 and C242) coordinate a zinc atom and are indispensable for maintaining p53 structure and function [18]. Cysteine is a highly conserved amino acid due to its function in catalytic centers and structural disulfides. On the other hand, the reactivity of the cysteine thiol group can also be detrimental to protein function when non-functional cysteines are acquired during evolution, especially on the surface of proteins, and these tend to be rapidly lost again. Paradoxically, cysteine has therefore been suggested to be both one of the most and one of the least conserved amino acids [23]. Although evolutionary conservation cannot unequivocally predict whether cysteines are functional or not, it is plausible that surface-exposed cysteines that are not conserved are dispensable for or even hamper protein function. We have previously shown that evolutionary acquisition and conservation of surface-exposed cysteines can be predictive of a functional role for a certain cysteine in redox signaling [24], and we thus analyzed the conservation of the cysteines in paralogs as well as vertebrate orthologs of human p53 (Figure 2B,C and Appendix A). In line with the strong conservation of functional cysteines, the Zn-coordinating cysteine homologous to human p53 C176, C238 and C242 are indeed present in all homologs of vertebrate species for which sequences were available, as well in the human p53 paralogs p63 and p73 (Figure 2B,C). Of the non-Zn-binding cysteines, those at positions 135, 141, 275 and 277 are also highly conserved in the human paralogs and vertebrate orthologs (C141 slightly less in Fish species). Conversely, acquisition without fixation seems to be the case for the surface exposed C182 and C229, and to a lesser extent for C124. These three cysteines are not conserved in human p63 and p73, and variation in the amino acid in these homologous positions can be observed in p53 in many vertebrates including mammals.

We aimed to connect the evolutionary conservation of cysteines in p53 to its function by testing the ability of various p53 Cys to Ser and Ala mutants to induce p21 expression in p53-deficient H1299 cells (Figure 2D and Appendix A). The ability if these mutants to form disulfide-dependent complexes was also assessed (Figure 2E and Appendix A). As expected, mutation of the Zn-finger cysteines (e.g., C176) leads to inactivation of the protein and loss of p21 induction (Figure 2D). This unfolded mutant forms many random disulfides as judged by the smear on non-reducing SDS-PAGE followed by Western blot (Appendix A). Mutation of the non-conserved C124, C182 and C229 did not affect p53-induced p21 expression, in line with the apparent flexibility for having a cysteine at these positions or not (Figure 2D and Appendix A). These mutants also did not migrate differently as compared with wild type under non-reducing conditions. The activity of C135S and C141S is completely or largely impaired (Figure 2D, whereas C135A and C141A induce p21 expression similar to p53 WT (Appendix A), possibly because alanine is a better substitute in terms of the local chemical environment of these cysteines. The observed smear on SDS-PAGE under non-reducing conditions for the C135S and C141S is indeed similar to that of the Zn-finger mutant C176S (Figure 2D and Appendix A), and could be an indication for unfolding and random intermolecular disulfide formation. The C135A and C141A mutants on the other hand migrated extremely similar to wildtype under non-reducing conditions. Mutation of the highly conserved Cys275 to either Ser or Ala impaired transcriptional activity but did not show signs of altered folding or differential S-S-dependent oligomerization (Figure 2B and Appendix A). This suggests that the C275 residue is essential for p53 transcriptional activity without grossly affecting its structure or redox sensitivity. C275 is in close contact with the DNA, which may leave little room for side-chain substitution. p53 C277 mutants (either to Ser or Ala) induced similar levels of p21 as WT p53 did, but lost the majority of redox-dependent high-molecular weight complexes on non-reducing SDS-PAGE in both H1299 cells and HEK293T cells (Figure 2D,E and Appendix A), suggesting that C277 partakes in redox-dependent intermolecular disulfides. Because C182 has previously been suggested to be redox sensitive, we also tested a C182S,C277S double mutant, which behaved extremely similar to the C277S and C277A mutants in terms of activity and redox-dependent intermolecular disulfide formation (Appendix A).

Taken together, p53 forms reversible intermolecular disulfides involving C277 under oxidizing conditions. 

### 3.3. Identification of p53 Disulfide-Dependent Binding Partners by MS

We set out to identify the disulfide-dependent interaction partners of p53. Using co-IP experiments with differentially tagged p53 constructs, we first excluded the formation of disulfide-dependent p53 dimers or oligomers (Appendix A). 

We performed quantitative LC-MS/MS to identify candidate disulfide-dependent binding partners of p53 by comparing the interactome of WT p53 and two cysteine mutants (C182S and C182S,C277S) (Figure 3A). Sample quality assessment showed that the expression and pull-down of the Flag-p53 proteins, as well as the (absence of) induction of intermolecular disulfide-dependent complexes were highly reproducible over four biological replicates (Appendix A). Overall raw MS data regarding the amount and intensity of identified proteins were also comparable between replicates (Appendix A). After filtering out proteins with less than two peptides, 1889 proteins in total were identified, out of which 162 proteins were significantly enriched in the Flag-p53 WT pull down upon diamide treatment (Log_2_ fold change >1 and adjusted *p*-value < 0.01). These included several proteins involved in redox signaling such as Trx and PRDX family-members (Figure 3B). Comparison of the proteins pulled down after diamide treatment with WT p53 versus C182S showed no significant changes in binding upon mutation of C182 (Figure 3C), consistent with our observations in the non-reducing SDS-PAGE and Western blot experiments (Figure 2B). The C182S,C277S double mutant on the other hand showed far less significant binders as compared with WT p53 or C182S, suggesting that C277 is required for many (but not all) of the intermolecular disulfide-dependent interactions of p53 (Figure 3D,E). A note of caution should be sounded here, because it cannot be excluded that the p53 C277S behaves differently than the C182S,C277S double mutant used in the screen. C277-dependent interaction partners of p53 need therefore be validated using the p53 C277S single mutant. Out of the 162 diamide-induced interacting proteins, 19 proteins were dependent on C277, including several well-known p53-binding proteins (for a list, see Appendix A).

Gene Ontology (GO) analysis for 162 diamide-induced p53 interactors (Appendix A) showed enrichment for several GO biological process terms related to the regulation of gene expression (Appendix A). From the perspective of GO molecular function, these binding partners were significantly associated with ‘protein binding’, but also with several terms related to disulfide oxidoreductase activity. Not surprisingly, the GO molecular function terms ‘p53 binding’ and ‘antioxidant activity’ were also significantly enriched among the binding partners (Appendix A). The enriched GO cellular component terms point at a function in the nucleus, which can be expected for transcription factor binding partners (Appendix A). 

### 3.4. C277-Dependent Interactions of p53 with 14-3-3θ and 53BP1

We were intrigued to find that the binding of a number of well-known p53 interactors and regulators was also affected by diamide treatment and depended on C277. These included (TP53BP1, MDM2, PSME3 and 14-3-3 family members (e.g., encoded by YWHAQ/E genes) [25,26,27] (Figure 3). We therefore decided to focus initially on the further validation of these binding partners and exploration of how these could affect p53 function, depending on C277 oxidation. 

14-3-3θ (also known as 14-3-3τ, shown in the volcano plot by its gene name YWHAQ), is the most significant hit identified to bind to p53 through C277 upon diamide in MS data (Figure 3). The LFQ data of the individual biological replicates show the reproducibility of this observation (Figure 4A). Note that this protein indeed also is found to bind p53 without diamide treatment as previously described [28], but with far less intensity as compared with the binding upon diamide treatment. Immunoprecipitation followed by non-reducing and reducing SDS-PAGE and Western blotting confirmed that the diamide-induced p53/14-3-3θ interaction was mediated by C277 (Figure 4B). The redox-dependent interaction between Flag-p53 and 14-3-3θ results in a large molecular weight band of over 100 kDa in the non-reducing gel, that migrates as monomeric Flag-p53 (about 55 kDa) and 14-3-3θ (about 28 kDa) upon reduction. This shows that the complex indeed is held together by an intermolecular disulfide involving p53 C277 (Figure 4B). RNAi-mediated knockdown confirmed that the shifted band indeed contains 14-3-3θ (Appendix A). The pattern of other intermolecular disulfide-dependent complexes containing Flag-p53 seemed not to be affected by 14-3-3θ knockdown, suggesting that this scaffold protein [29] does not mediate the other redox-dependent interactions of Flag-p53, for instance by forming complexes containing multiple disulfides (Appendix A).

Diamide-induced and C277-dependent binding of 53BP1 was also validated. Figure 4C shows the reproducibility of the LFQ data of the individual replicates for each condition. Note that without diamide treatment 53BP1 is not identified, and that the data for this condition represent identical imputed values (set to value 20 (Log2), which is around the lowest value in the whole dataset). Both endogenous 53BP1 and overexpressed GFP-53BP1 can be co-immunoprecipitated by Flag or HA tagged p53. Paralleling reducing and non-reducing SDS-PAGE followed by Western blot shows that p53 and 53BP1 indeed form an intermolecular disulfide-dependent complex involving p53 C277 (Figure 4D and Appendix A). Without diamide treatment we observed extremely little 53BP1 binding to p53, although this protein–protein interaction has been extensively studied by others without the use of oxidizing conditions [30]. An explanation for this apparent discrepancy could lie in the sample preparation that we use for the identification of disulfide-dependent interactors. The immunoprecipitation protocol involves a high-salt wash (1 M NaCl) in order to lower the number of non-covalent binders. Indeed, milder washing conditions reveal 53BP1 binding also in the absence of diamide treatment, be it with far lower abundance (Appendix A). In the absence of diamide, part of the bound GFP-53BP1 is visible as an intermolecular disulfide-dependent complex, suggesting that this disulfide can form under endogenous conditions. The GFP-53BP1 fraction that is pulled down with p53 that does not migrate as an intermolecular disulfide-dependent complex, and hence binds p53 only through non-covalent interactions [31] was affected the most by the high salt wash (Appendix A). Interestingly, both the disulfide-dependent and -independent interaction increase dramatically in WT cells upon diamide treatment, which could mean that part of the disulfide-stabilized p53 and GFP-53BP1 complex is reduced during sample preparation, while maintaining the interaction. Accordingly, far less disulfide-independent p53-53BP1 binding is pulled down by the p53 C277S mutant. A small amount of disulfide-dependent p53-S-S-53BP1 complex is observed upon washing with low-salt buffer, which disappears upon a high-salt wash. This observation can be explained by the pull down of endogenous WT p53-S-S-GFP-53BP1 complexes with HA-p53 C277S in a non-covalent manner. 

### 3.5. p53 C277 Is Dispensable for the p53-Dependent Response to Nutlin-3a, Oxidant Treatment and DNA Damage

Disulfide-dependent binding of regulatory proteins could affect the transcriptional activity of p53 through for instance altered subcellular localization, (de)stabilization or differential target promoter binding. To test this, we devised a doxycycline (Dox)-inducible system expressing WT or C277S in p53 KO RPE^Tert^ cells (Figure 5A). The localization of p53 WT and C277S was highly similar and both were mainly present in the nucleus in these cells (Figure 5B). After 48 h of Dox addition, we tracked p53 protein levels and activity upon Nutlin-3a treatment, diamide (Figure 5C–E) and the DNA damaging agent NCS (Figure 5F–H), and we found that both WT and C277S were stabilized and activated to a similar extent in response to these compounds. Diamide treatment also did not affect the response to Nutlin-3a (Figure 5C–E). (Poly) ubiquitination of both WT and C277S was blocked to a similar extent in response to both diamide and H_2_O_2_ (Figure 5I,J). 

## 4. Discussion

Oxidation of protein cysteine thiols leads to a suite of PTMs that can reversibly alter protein structure and function. In this way, cysteine oxidation-dependent redox signaling regulates a variety of biological processes including cell proliferation, differentiation, migration and regeneration [32,33,34]. The methods used to detect reversible protein oxidation are in general based on differential alkylation of cysteines prior and post reduction, and hence the type of reversible oxidation (e.g., sulfenic acid, *S*-glutathionylation, sulfenamide, inter- or intramolecular disulfide) is lost in this process. Strategies using sequential reduction in specific oxidative PTMs have been used to discriminate for instance proteome-wide *S-*GSHylation and *S*-nitrosylation [35] by MS/MS. However, no method exists to date to identify or distinguish intra- or intermolecular disulfides in a proteome-wide manner. The amount of theoretically possible tryptic digests containing peptides from two distinct proteins and an intact disulfide is virtually endless. Intermolecular disulfides can be identified for a protein of interest by first comparing the interactomes of the wildtype protein and a cysteine mutant and subsequently test whether the protein of interest and a cysteine-dependent interactor indeed migrate as a reduction-sensitive complex on SDS-PAGE under non-reducing conditions [5]. The tumor suppressor p53 had already been reported to undergo reversible cysteine oxidation in response to oxidizing agents both in vitro and at endogenous levels in live cells [8], but the nature of the reversible oxidation remained elusive in that study. In the present study, we provide evidence that cysteine oxidation of p53 leads to the formation of several intermolecular disulfide-dependent complexes, most of which depend on C277. This cysteine is also implicated in the binding to cysteine-directed covalent drugs aimed at refolding mutant p53 [36], which means that these compounds could also likely interfere with the intermolecular disulfide-dependent, p53-containing complexes described in this study. Note that the presented MS screen compares the binding of proteins to WT p53 with and without diamide treatment to binders of the C182S and C182S,C277S mutants in the presence of diamide. No changes were observed comparing WT p53 and C182S, whereas many proteins did not bind the C182S,C277S double mutant, but based on the screen we cannot exclude that proteins can bind to either C182 or C277. It is therefore important to validate hits from the MS screen by other means using the single p53 C182S and p53 C277S mutants as well. Similarly, for practical and financial reasons the interactome of p53 cysteine mutants without diamide treatment was not assessed, and validation experiments should include both treated and untreated samples. 

A number of disulfide-dependent and validated hits from our MS screen are known interactors and regulators of p53, including MDM2, 53BP1 and 14-3-3. However, the original studies describing the interactions of these proteins with p53 did not study redox or cysteine dependency [28,30,37]. We show that at least for 53BP1 the interaction with p53 is indeed not strictly dependent on the disulfide (Appendix A). This experiment shows that the disulfide stabilizes the interaction in the co-immunoprecipitation assay and makes it resistant to a stringent high-salt wash. It remains to be seen to what extent this translates in the in vivo situation, and whether this means that in cells the strength or duration of the p53-53BP1 protein–protein interaction is also significantly enhanced as compared with a purely electrostatic interaction upon disulfide formation. The observation that the C277S mutant can still interact with 53BP1 suggests that the p53-53BP1 interaction occurs prior to oxidation, and that the disulfide forms between two cysteines that are already in close proximity. This could be a general concept for the formation of intermolecular disulfides, and potentially explains why p53 does not form random intermolecular disulfides with a wide range of proteins. The disulfide could in this case either strengthen a functional protein–protein interaction or lead to a conformational change that alters the protein–protein interaction in such a way that it interferes with its function. If the latter were the case, one might predict that C277 would maybe not be conserved, similar to C182 and C229, whereas it displays strong evolutionary conservation. On the other hand, if the disulfide-dependent interaction would greatly enhance the regulatory function of an interaction partner we would expect to have observed differences in the transcriptional activity or stability in the C277S mutant, which we did not. The latter might be because multiple proteins with opposing regulatory functions for p53 seem to interact with C277. Since both the C277S and C277A mutants still have transcriptional activity, we can conclude that cysteine oxidation is not absolutely required for p53 function, but that it could maybe provide a means for fine-tuning target selection or the duration of a regulatory response. An alternative function for disulfide formation could be to prevent irreversible oxidation of cysteines in p53, although as far as we know there is no evidence that this actually occurs.

We previously showed that diamide (but not peroxide) -mediated oxidizing conditions induce p53 stabilization and activation through p38MAPK-dependent signaling, which was independent of surface-exposed p53 cysteines (including C277) [38]. p38MAPK-dependent p53 activation under oxidizing conditions could therefore obscure the effects of disulfide-dependent binding partners under the conditions tested. Furthermore, the majority of WT p53 is still reduced upon diamide treatment, and this could conceal potential regulatory effects of the disulfide-dependent interactions which have a relatively low stoichiometry. 

Interestingly, p53 C277 mutations have been identified in several human tumor tissues [39,40]. However, since Cys277 is in the DNA-binding domain and actually is in contact with the DNA [41], it would be difficult to distinguish whether these mutations (the majority of which are changes to large hydrophobic residues) would contribute to oncogenic transformation because of loss of protein–protein interactions or because of altered DNA binding.

Although in the present study we aimed to identify a potential functional role for redox regulation of p53, it might also be that the functional consequence lies ‘at the other end’ of the intermolecular disulfides. The disulfide-mediated p53-53BP1 interaction may for instance alter the efficiency of 53BP1-dependent non-homologous end joining in DNA-damage repair. Likewise, locking p53 to MDM2 may interfere with the ubiquitination-dependent breakdown of MDM2 substrates other than p53. The 14-3-3 proteins also have many more binding partners besides p53, and the intermolecular disulfide-dependent interaction may alter its adaptor function towards other proteins. We have previously shown that the FOXO transcription factors do not bind 14-3-3 proteins in a cysteine-dependent manner [42], suggesting that not all 14-3-3 interactors bind in a redox-dependent manner. To what extent covalent binding of p53 to these proteins will affect their function depends of course on the stoichiometry of the interaction.

The intrinsic sensitivity for oxidation of cysteine thiols in proteins depends on a number of variables including their pKa, solvent accessibility and local protein folding. Reactivity to H_2_O_2_, for instance, can vary between several orders of magnitude. It has therefore been proposed that within live cells, oxidation of most cysteines by relatively low levels of H_2_O_2_ probably occurs indirectly, for instance, catalyzed by peroxiredoxins [12,43]. In this study, we observed that disulfide-containing complexes of p53 were only detectable in response to diamide but not H_2_O_2_ treatment when high glucose media was used. We have previously shown that intermolecular disulfides can be detected in the human 2-Cys peroxiredoxins [12] as well as the FOXO3 and FOXO4 transcription factors [24,42,44] upon treatment of cells with H_2_O_2_, starting even at much lower concentrations and cultured in high glucose media. The work by Held et al. showed before that cysteines in endogenous p53 are oxidized by diamide and not by H_2_O_2_. Furthermore, the diamide-induced oxidation of p53 in live cells occurs at much lower concentrations as compared in vitro on recombinant p53 [8]. These observations suggest that in live cells, p53 cysteine oxidation also does not occur directly or requires an additional factor or catalyst. Glutathione is, due to its abundance, the most likely direct target of the thiol oxidant diamide, and it might be that in cells, p53 oxidation is mediated by oxidized glutathione, but future work is needed to explore this idea. Alternatively, diamide-induced inhibition of (GSH-dependent) disulfide reduction could expose the continuous turnover of intermolecular disulfides that form between p53 and interacting proteins. The latter could be in line with the observation that under low glucose culturing conditions, or when continuously produced by the addition of glucose oxidase to the media, H_2_O_2_ does lead to detectable p53 containing intermolecular disulfides. Glucose, through the pentose phosphate pathway, drives production of NADPH which is required for both the GSH- and Trx-dependent disulfide reduction systems. The question remains why inhibition of the reductive system is needed to expose the formation of disulfides upon H_2_O_2_ treatment for some proteins (e.g., p53) but not others (e.g., FOXOs). In any case, a differential pattern of cysteine oxidation in response to different oxidants is an example of specificity in redox signaling, and it is not unthinkable that a differential cellular response is required upon oxidizing conditions induced by more oxidants (i.e., H_2_O_2_) or by lower reductive power (i.e., diamide or Auranofin). 

## 5. Conclusions

Taken together, here we show that cysteine oxidation of p53 can come in the form of intermolecular disulfides involving a large but defined set of binding partners. Future studies are needed to understand their functional importance in the context of normal physiology and tumor biology.

## Figures and Tables

**Figure 1 antioxidants-10-01578-f001:**
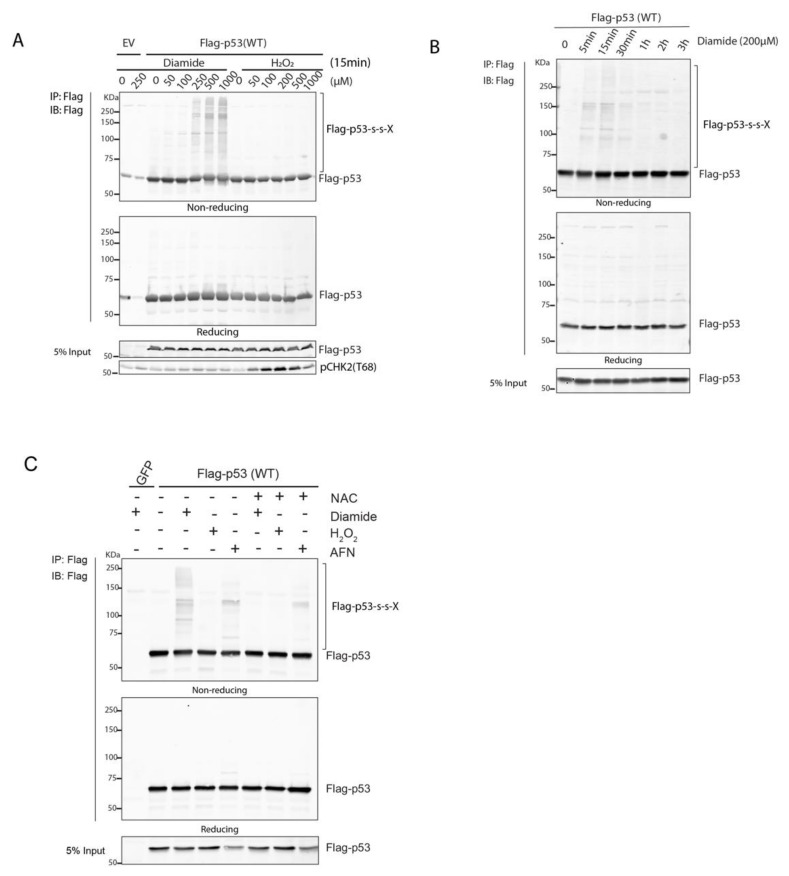
p53 forms redox-sensitive protein–protein interactions upon oxidant treatment. (**A**) Diamide, but not H_2_O_2_, induces redox-dependent interactions between p53 and other proteins in a dose-dependent manner. Flag-p53 immunoprecipitated from diamide-treated HEK293T cells migrates in several high-molecular weight bands under non-reducing conditions. pChk2 levels indicate that H_2_O_2_, but not diamide, induces an ATM-dependent DNA damage response. (**B**) Diamide-induced p53 complexes are reversible and resolved over time by the cellular antioxidant system. (**C**) AFN treatment also induces the formation of p53 complexes, be it with a pattern distinct from diamide. NAC pre-treatment prevents most diamide-induced complex formation.

**Figure 2 antioxidants-10-01578-f002:**
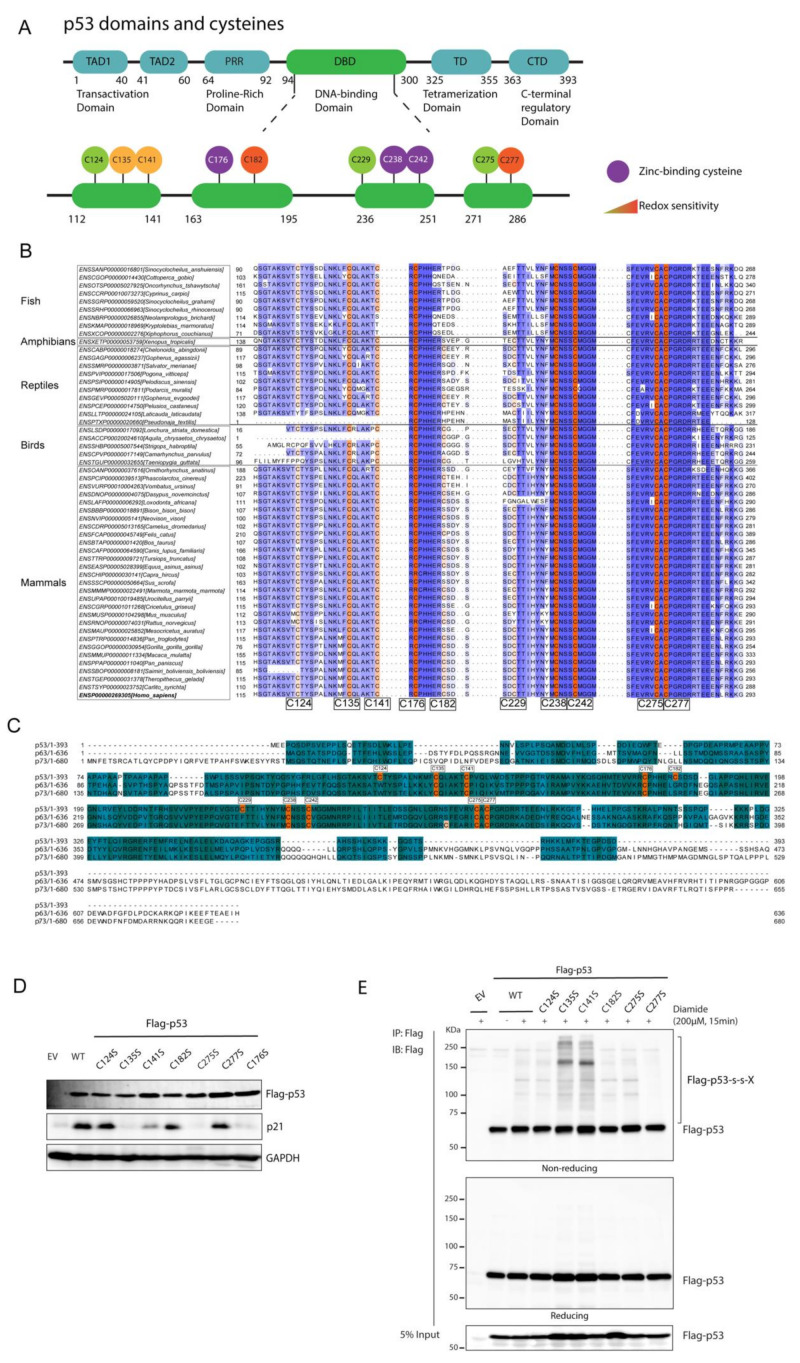
p53 forms disulfide-dependent protein–protein interactions through cysteine 277. (**A**) Scheme of domains and cysteines of the p53 protein. The p53 protein contains two Transactivation Domains (TAD1 and TAD2), a Proline-Rich Domain (PRR), a DNA-binding Domain (DBD), Tetramerization Domain (TD) and a C-terminal regulatory Domain (CTD). All ten cysteines of p53 are located within its DBD. C176, C238 and C242 (in purple) are Zinc-coordinating cysteines and are essential for maintaining p53 structure. C176, C182, C229, C242 and C277 are surface-exposed cysteines. C135, C141, C182 and C277 have been shown to be prone to oxidation (in yellow and orange). The locations of the p53 protein domains and regions are adapted from the TP53 database (http://p53.fr/) (accessed on 30 January 2019). (**B**) Alignment of human p53 protein sequences from the five classes of vertebrate species (fish, amphibians, birds, reptiles and mammals). Only the conservation around the cysteines is shown. For a full alignment of the DBD see Appendix A. (**C**) Alignment of protein sequences of p53 and its paralogs p63 and p73 from *Homo Sapiens*. The alignments are colored based on the extent of conservation. Cysteines are indicated in orange. (**D**) The transcriptional activity of p53 cysteine mutants was assessed by the induction of p21 protein levels upon expression in p53-deficient H1299 cells. (**E**) C277 is required for several of the redox-dependent interactions of p53, as shown by Immunoprecipitation followed by SDS-PAGE and Western Blot under non-reducing conditions. The experiment was performed in H1299 cells (p53-deficient) that were transiently expressing p53 WT and cysteine mutants (to Serine). Note that the strong induction of disulfides in the C135S is also observed in the unfolded C176S mutant and is absent in the C135A and C141A mutants (see Appendix A).

**Figure 3 antioxidants-10-01578-f003:**
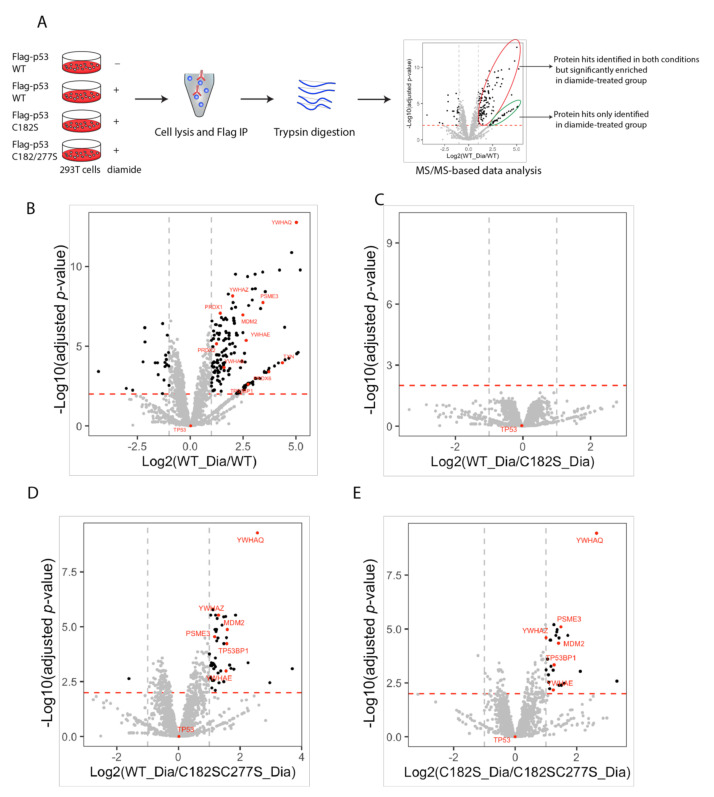
Identification of disulfide-dependent p53 interactors by MS/MS analysis. (**A**) Scheme for the identification of p53 disulfide-dependent interactors by MS/MS. HEK293T cells were transfected with Flag-p53 WT, C182S, and C182S,C277S. 48h after transfection, cells were treated with diamide, followed by immunoprecipitation and processing for MS/MS. MS data were further analyzed in R and plotted in a volcano plot. Protein hits with >2-fold enrichment and an adjusted *p*-value < 0.01, are considered significant interactors (black dots). Hits circled in red are identified in both conditions, whereas the green circle indicates proteins identified in only one of the samples. Data analysis was based on 4 biological replicates for each condition. (**B**) Volcano plot showing interactors of Flag-p53 WT with and without diamide treatment. (**C**) Volcano plot showing interactors of Flag-p53 WT vs. C182S, both with diamide treatment. (**D**) Volcano plot showing interactors of Flag-p53 WT vs. C182S,C277S, both with diamide treatment. (**E**) Volcano plot showing interactors of Flag-p53 C182S vs. C182S,C277S, both with diamide treatment.

**Figure 4 antioxidants-10-01578-f004:**
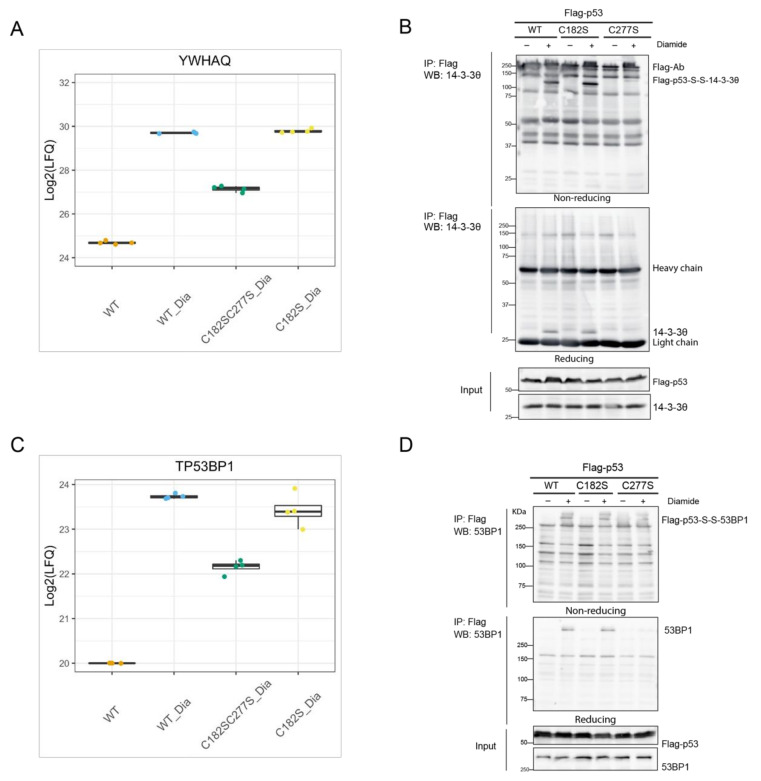
Validation of disulfide-dependent binding partners of p53. (**A**) Comparison of the LFQ values of the YWHAQ gene product (encoding the 14-3-3θ protein) in different conditions. (**B**) Validation of the disulfide-dependent interaction between p53 and 14-3-3θ by IP followed by WB. (**C**) Comparison of the LFQ values of the TP53BP1 gene product (encoding the 53BP1 protein) in different conditions. Note that the TP53BP1 gene product was not found in any of the replicates of the untreated condition (WT) and a missing value (NA) was observed upon Log2 transformation. This was manually imputed by a value of 20(Log2) (near to the lowest value in the whole dataset) during data analysis. (**D**) Validation of the disulfide-dependent interaction between p53 and 53BP1 by WB.

**Figure 5 antioxidants-10-01578-f005:**
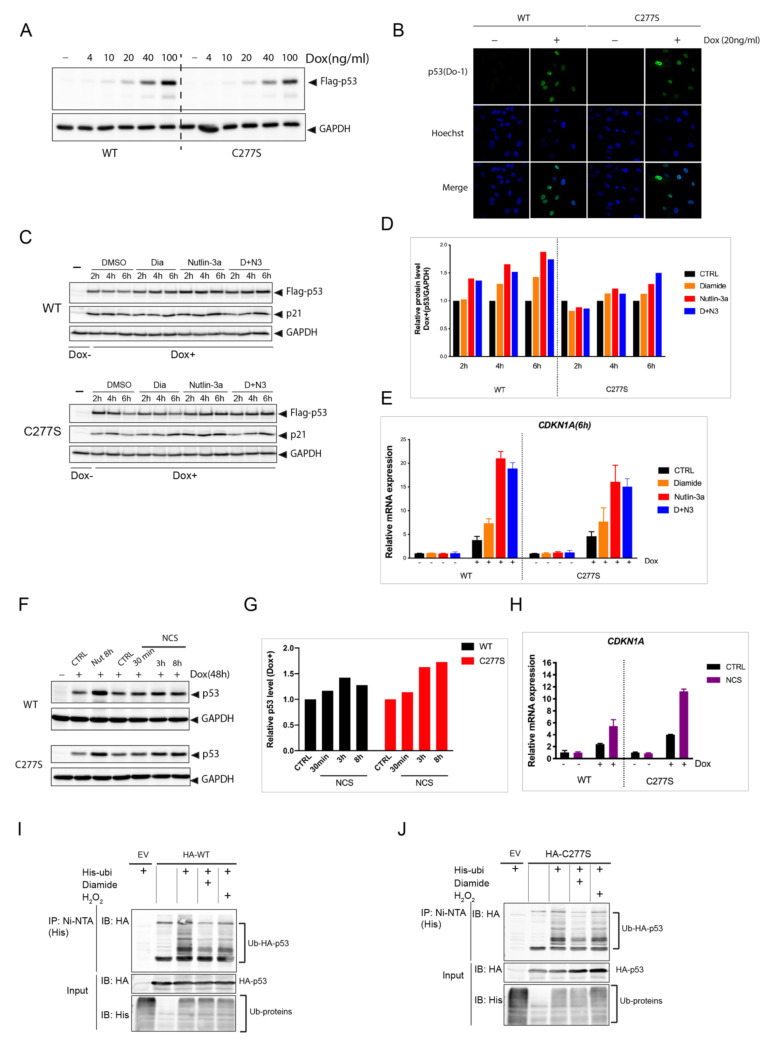
p53C277S is stabilized and activated similar to p53 WT in response to Nutlin-3a, oxidant treatment and DNA damage. (**A**) Dox-inducible expression of Flag-p53 WT and C277S in p53 KO RPE^Tert^ cells upon titration with doxycycline. (**B**) Immunofluorescence images showing the sub-cellular localization of Flag-p53 WT and C277S in dox-inducible p53 KO RPE^Tert^ cells. Both are mainly localized in the nucleus. (**C**) Flag-p53 level in dox-inducible expressing p53 WT and p53C277S RPE^Tert^ cells upon different stimuli. Flag-p53 WT and C277S were induced by Dox (20ng/ml, 48h) in p53 KO RPE^Tert^ cells, followed by treatment with diamide (200 μM), Nutlin-3a (10 μM), or both (D+N3) for the indicated time. Total cell lysate was harvested and the levels of Flag-p53 WT, C277S, p21 and GAPDH (as a loading control) were evaluated. The blots are representative for the results of at least three independent experiments. (**D**) Quantification of Flag-p53 WT and C277S protein levels relative to GAPDH from (C). (**E**) *CDKN1A* (p21) mRNA expression in Dox-inducible expressing p53 WT and C277S RPE^Tert^ cells upon different treatments as determined by qPCR. (**F**) Flag-p53 level in the Dox-inducible expressing p53 WT and C277S RPE^Tert^ cells upon Neocarzinostatin (NCS)(250 ng/ml) or Nutlin-3 treatment. (**G**) Quantification of Flag-p53 WT and C277S protein levels relative to GAPDH from (F). (**H**) *CDKN1A* (p21) mRNA expression in Dox-inducible expressing p53 WT and C277S RPE ^Tert^ cells upon NCS treatment as determined by qPCR. Ubiquitination of p53 WT (**I**) and C277S (**J**) upon diamide or H_2_O_2_ treatment. HEK293T cells transiently expressing HA-p53 (WT or C277S) and His-ubiquitin were treated with diamide (200 μM) or H_2_O_2_ (200 μM) for 15 min. His-tag-Ubiquitinated proteins were precipitated using Ni-NTA agarose beads and analyzed by Western blot using His or HA antibodies.

## Data Availability

Data is contained within the article and supplementary materials. The raw data presented in this study have been deposited into the ProteomeXchange via the PRIDE database with identifier PXD026893.

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
