# Peer review of "p53 Forms Redox-Dependent Protein–Protein Interactions through Cysteine 277"

_antioxidants, 2021, doi:10.3390/antiox10101578_

Round 1

Reviewer 1 Report

Formation of reversible intramolecular disulfide bounds between cysteine residues is known to play a central role in redox signaling, controlling many protein activities. However, the importance of such covalent bounds in protein-protein interactions, between partners of a complex that could stabilize the protein-protein interaction, is much less documented. In this manuscript, Shi et al. addressed the functional importance of p53 to form intermolecular disulfide bridges with interacting proteins. The transcription factor p53 is a key player in cellular stress response, and its function is controlled by numerous reversible post-translational modifications depending on the nature of the stress. To investigate the intermolecular disulfide bound formation in p53, they performed co-immunoprecipitation experiments under different redox status or using different p53 Cys mutants and identified redox sensitive partners using quantitative mass spectrometry and Western blot analyses. As previously reported, it was shown that this disulfide complexes formation is linked to diamide-dependent oxidation, but not H2O2 induced-oxidative stress. These experiments characterized the importance of one particular conserved cysteine residue of p53, Cys 277, to form disulfide links with many proteins, including redox controlling enzymes and two well-known p53-binding proteins, 14-3-3 family members and 53BP1. Nevertheless, the p53 Cys 277 mutant appeared to be still transcriptionaly active: p53 C277S or A mutants induced similar level of p21 expression as WT, and both WT and C277S were similarly stabilized and activated in response to Nutlin-3 treatment, oxidative signaling or DNA damage. This indicated that the identified disulfide-dependent interactors are not critical for p53 activity.

From this, it was concluded that the complex-dependent C277oxidation does not represent an on/off switch for p53 transcriptional activity and it was proposed that the formation of the intermolecular disulfide bound forms after complex formation, strengthening the protein-protein interaction. This will provide a means for better controlling p53 activity by fine-tuning target selection or duration of the regulatory response. Alternative hypotheses are discussed.

Major comment

Line 282, the explanation for the behavior difference between mutants C135S-C141S and C135A-C141A is not convincing, since hydrophobicity is not the problem here for this solvent exposed residue, electrophilicity is probably a more critical parameter.

Fig S3A, the annotations on the top of this Figure are confusing. What is the difference between the two last lanes (Flag-p53+HA-p53) since the oxidized-dependent complex around 100 kDA is only present in the very last lane? Was diamide treatment applied in the two conditions? Similarly, isn’t it unexpected to detect the Flag-p53 complex in the second lane in absence of diamide treatment?

The choice of the mutant used in co-IP experiment is unexpected and might lead to inappropriate conclusions. What is the purpose to use the C182SC277S mutant instead of the simple C277S mutant? Because of this, the conclusion that “out of the 162 diamide-induced interacting proteins, 19 proteins were dependent on C277” is erroneous, since C182S was also mutated in the IP construct. Can it be excluded that C182 is not active per se but that its activity is potentiated by C277S mutation? Experiments using simple C277S mutation should be included or, at least, the use of the double mutant needs to be justified.

Is it possible that formation of the covalent interaction protects Cys 277 of irreversible oxidation, preventing p53 controlled inactivation?

Minor comments

Something missing in lines 254-255

What is C134S mutant in lines 281-284? I guess one should read C135S.

In conclusion, this manuscript describes very interesting observations and an interesting discussion. However, it might be improved before publication.

Reviewer 2 Report

The authors of "p53 forms redox-dependent protein-protein interactions through cysteine 277" study the oxidation of cysteine residues found in p53 and how this process affects the interaction of p53 with its partners. Intriguingly, Shi et al. could show that a specific residue, C277, is responsible for protein interactions via S-S-bridges after oxidation. Their findings extend the current knowledge about p53 and how its functions can be (fine) tuned after oxidative stress. Nevertheless, the authors need to address several points before publication.

General comments

1.) The results and discussion sections are very mixed, especially for chapter 3.6. The authors should separate results and discussion section more stringent in their revised version.

2.) The oxidation of cysteines and the formation of complexes between p53 and other proteins can only be detected after diamide treatment, but not after H2O2 treatment. Diamide treatment is though not a physiologic condition for cells and the question remains if the results shown here do have a high translational relevance. Since, H2O2 quickly is degraded (within a few minutes) after addition into cell culture media, the authors may try to incubate cells with H2O2 in PBS or use the more stable t-butyl-hydrogen peroxide for their experiments.

3.) Although not being a hot-spot mutation, mutations of C277 do occur in human tumors. Is there knowledge about loss of p53-C277 interactions in these malignancies? The author may discuss this issue.

4.) The authors need to explain their data/figures better (see my specific comments).

Specific comments

1.) In Fig. 2 and S2 the authors show the interaction patterns of different p53 mutants after diamide treatment. However, they do not mention the C182, C229 and C124 mutants in the corresponding paragraph (3.2) although they are shown in the figures. Especially, the C182 mutant(s) are important and used for MS analysis.

2.) The authors used for MS analysis the C182/C277 double mutant and thus, should perform similar experiments as shown in Fig. 2 and S2 with the double mutant, too.

3.) I highly recommend combining Fig. 2 and 6, since the authors start at the end of their results sections to describe Fig. 2 again in relation to the evolutionary conservation of these cysteins in p53, which makes the manuscript difficult to read.

4.) What was the rationale to exclude the C277 mutant for MS analysis? The authors only used the WT p53, p53-C182 and p53-C182C277. Why did the authors only study the p53 mutants after diamide treatment and not without? See also connected to this my comments in the next section below.

5.) The authors describe at the end of paragraph 3.5 the stabilization of p53 via the p38MAPK signaling. However, they do not refer to a figure in their manuscript, which is presumably Fig. 5F-H and do not explain how they modulated p38MAPK pathways.

Specific comments to the supplement

1.) In Fig. S2A the Western Blot is highly cut and re-assembled. The authors should show their results on one connected membrane instead.

2.) The label of Fig. S3A appears to be wrong when looking at the expression and interaction profile of the p53 constructs.

3.) In Fig. S4C and S4D it is obvious that 1 out of 4 replicates from the WT-Dia group has a strong outlier. Was this samples included in the following analysis? If though, this might have strongly affected their conclusions drawn here. It the outlier is not included all diamide treated groups/p53 mutants appear quite similar in the PCA analysis (Fig. S4D).

4.) Although the authors shown in the lowest panel of Fig. S6 that 14-3-3 protein level could be reduced using siRNA, the upper panels, especially the left one, do not show any effect of si14-3-3-KD. Thus, the KD either did not work in the upper panels or has no effect.

Author Response

Reviewer 2

The authors of "p53 forms redox-dependent protein-protein interactions through cysteine 277" study the oxidation of cysteine residues found in p53 and how this process affects the interaction of p53 with its partners. Intriguingly, Shi et al. could show that a specific residue, C277, is responsible for protein interactions via S-S-bridges after oxidation. Their findings extend the current knowledge about p53 and how its functions can be (fine) tuned after oxidative stress. Nevertheless, the authors need to address several points before publication.

General comments

1.) The results and discussion sections are very mixed, especially for chapter 3.6. The authors should separate results and discussion section more stringent in their revised version.

We thank the reviewer for this suggestion. In our opinion it can improve legibility to briefly discuss data in the results section, for instance to clarify the use of certain methods or the rationale behind follow up experiments. We have gone over the manuscript and deleted some points that might have been too extensively discussed in the results section. We have now combined figure 6 and figure 2 (new figure 2) as per the reviewer’s very good suggestion below. The new figure aims to connect evolutionary conservation to functional studies and we think it would not make sense to only describe the direct result without discussing the connections between conservation and function.

2.) The oxidation of cysteines and the formation of complexes between p53 and other proteins can only be detected after diamide treatment, but not after H2O2 treatment. Diamide treatment is though not a physiologic condition for cells and the question remains if the results shown here do have a high translational relevance. Since, H2O2 quickly is degraded (within a few minutes) after addition into cell culture media, the authors may try to incubate cells with H2O2 in PBS or use the more stable t-butyl-hydrogen peroxide for their experiments.

Like the reviewer we were also puzzled by the absence of disulfide formation by H2O2. After all, although mechanistically distinct, both diamide and H2O2 results in oxidized GSH and loss of reductive capacity. The low abundance of p53 cysteines as compared to other thiols including GSH makes it unlikely that diamide at the used concentrations reacts with p53 directly in live cells. Furthermore, earlier work from our group on the FOXO transcription factors using similar conditions in terms of H2O2 concentration and treatment duration did show S-S-dependent heterodimerization of FOXO transcription factors (Dansen, 2009; Putker, 2013; Putker 2015). We think that rapid degradation of H2O2 is likely not the explanation for the absence of detectable p53 oxidation, because at the used concentrations we get a clear induction of the DNA damage response, meaning that H2O2 does get into the cell (and probably the nucleus).

Based on the reviewer’s excellent suggestions we have now looked further into this. Treatment with H2O2 in PBS did not result in p53 S-S-dependent heterodimerization, indeed ruling out that rapid decomposition of H2O2 by media components is at the basis of the difference with diamide. However, when the HEK293T cells are cultured in low glucose (1 g/l) instead of the high glucose media (4.5 g/l) that we normally use, we now do observe p53 S-S-dependent heterodimerization, and with a similar pattern on non-reducing SDS-PAGE/WB as compared to diamide treatment. Glucose is an important source of NADPH, which is the main driver of the cellular GSH and Trx dependent reductive systems. We therefore speculate that under high glucose conditions the p53 S-S-dependent heterodimers are rapidly reduced. The observation that diamide-induced S-S-dependent heterodimerization of p53 is not turned over equally fast in high glucose media could possibly be explained by direct inhibition of the reductive system by diamide. We have now also used enzymatic production using Glucose Oxidase to the culture media to induce continuous H2O2 production. This approach indeed also leads to p53 S-S-dependent heterodimerization, albeit to a lesser extent than diamide treatment or bolus H2O2 in low glucose conditions. Note that since the substrate for Glucose Oxidase is obviously glucose (which is also important for reduction of disulfides as outlined above), it is somewhat difficult to compare Glucose Oxidase treatment in low and high glucose media. The data have now been included in Figure S1.

We are very grateful to the reviewer for the suggestion to look more into the induction of p53 S-S-dependent heterodimers by H2O2. We agree that diamide is not a physiological compound (although its downstream product GSSG is), and being able to detect p53 oxidation also with H2O2 is reassuring that our results could have translational relevance. As to what the mechanism is behind the observation that the reductive capacity of cells cultured in high glucose is not sufficient to prevent diamide induced disulfides whereas it is capable of the rapid turnover/prevention of H2O2 induced disulfides is interesting but beyond the scope of this study.  We have updated the manuscript in light of the newly obtained data using H2O2 in low glucose media. Specifically, we have added a new supplementary figure (Figure S1) that shows the new experiments, and updated the results and discussion section. 

3.) Although not being a hot-spot mutation, mutations of C277 do occur in human tumors. Is there knowledge about loss of p53-C277 interactions in these malignancies? The author may discuss this issue.

Indeed, mutations in p53 Cys277 have been documented in the COSMIC database, and although these are indeed not hotspot mutations, they actually are relatively prevalent (Nguyen T, Human Mutation 2014). But since Cys277 is in the DNA-binding domain and actually is in contact with the DNA (Buzek, NAR 2002), it is difficult to distinguish whether these mutations (which most frequently are mutations to the large hydrophobic F, Y or W) would contribute to oncogenic transformation because of loss of protein-protein interactions or because of altered DNA binding. As far as we are aware, no systematic studies have been done to map altered protein-protein interactions in the context of tumors with p53-Cys277 mutations. We have added a few lines regarding mutations in C277 in cancer to the discussion section.

4.) The authors need to explain their data/figures better (see my specific comments).

We have revised the manuscript according to your comments as indicated below.

Specific comments

1.) In Fig. 2 and S2 the authors show the interaction patterns of different p53 mutants after diamide treatment. However, they do not mention the C182, C229 and C124 mutants in the corresponding paragraph (3.2) although they are shown in the figures. Especially, the C182 mutant(s) are important and used for MS analysis.

We have updated paragraph 3.2 to include the observations on the other cysteine mutants. Since we updated figure 2 according to this reviewer’s point 3 this paragraph has undergone major changes.

2.) The authors used for MS analysis the C182/C277 double mutant and thus, should perform similar experiments as shown in Fig. 2 and S2 with the double mutant, too.

Thank you for this suggestion. We did already perform that experiment but had not included it in the manuscript since there were no changes observed as compared to the C277S mutant. We agree that it is better to include this mutant in the manuscript, and it is now presented in Fig. S3A, B and Fig. S5A. 

3.) I highly recommend combining Fig. 2 and 6, since the authors start at the end of their results sections to describe Fig. 2 again in relation to the evolutionary conservation of these cysteines in p53, which makes the manuscript difficult to read.

We are grateful for this suggestion; it makes much more sense indeed. We have now combined Fig.2 and Fig.6 (now Fig.2) and rewritten the associated text in the revised manuscript.

4.) What was the rationale to exclude the C277 mutant for MS analysis? The authors only used the WT p53, p53-C182 and p53-C182C277.

Reviewer 1 had the same question; we have copied our response below:

We agree with the reviewer that it would have been better to include the C277S mutant also in the MS/MS screen for cysteine dependent p53 interactors. But at the time we performed the screen, our hypothesis was that Cys182 would be involved in the S-S-dependent p53 heterodimerization, since Cys182 was identified as the most sensitive to oxidation in cells by Held and coworkers (Held, Mol Cell Proteomics 2010). In Fig 5 of that paper, Cys277 is also shown as prone to oxidation by diamide, but to a lesser extent. We therefore included the C182S/C277S mutant in case proteins would start forming disulfides with C277 in the absence of C182 (similar to what the reviewer suggests). But then it turned out that we did not find any significant interactors that were dependent on C182, but many p53 WT interactors were absent in the C182S;Cys277S double mutant. We agree that combined these observations could mean either that these interactors bind exclusively to Cys277, or that they could start interacting with Cys182 in the absence of Cys277. We have tried to set up the mass spec screen again, now including the Cys277S mutant, but for some reason we had technical difficulties getting equal expression of this mutant compared to the other mutants in large scale experiments, which is important for generating high quality mass-spec data. We decided to continue with the original screen, since the technical quality of the MS data was very high. Besides, from our experience with MS screens we realized that any screen is just that: a screen. Targets will need to be validated anyways by other means. We therefore included the single mutants in Co-IP/Western blot experiments, and confirmed the strict dependency on Cys277 for several targets (see Fig 4). We have now also included the C182S;C277S double mutant in the IP experiments presented in Figs S3 and S5 as suggested by Reviewer 2, and the pattern of disulfide-dependent interactions looks very similar to that of the C277S single mutant. We will add a few sentences to the discussion explaining that we cannot exclude that some of the proteins we find to interact with WT p53 but not with p53C182S/C277S could in theory interact with C182 in the absence of Cy277, although we did not find different interactomes comparing WT and C182S.

Why did the authors only study the p53 mutants after diamide treatment and not without? See also connected to this my comments in the next section below.

We agree that in an ideal world it could be helpful to include also untreated mutants in the p53 interactome screen. Nevertheless, since we perform our screen in quadruplicate for reproducibility, the number of samples would rapidly grow out of hand if we would include an untreated control for each mutant. Including more variables not only is costly but also increases the chance of comparing samples of different quality or proteome depth. Furthermore, and similar to not having the C277S single mutant in the experiment: the MS experiment is ‘just’ a screen and hits need to be validated by for instance Co-IP/Western blot experiments, and in those smaller scale experiments it is easier (and much cheaper) to include more controls. Indeed, the experiments validating the hits 14-3-3 and 53BP1 presented in Figure 4 do include the C277S single mutant treated with and without diamide. We will add a few sentences to the discussion that hits need to be validated, including their dependence on oxidizing conditions.

5.) The authors describe at the end of paragraph 3.5 the stabilization of p53 via the p38MAPK signaling. However, they do not refer to a figure in their manuscript, which is presumably Fig. 5F-H and do not explain how they modulated p38MAPK pathways.

The statement regarding p38MAPK at the end of paragraph 3.5 does not relate to a figure but refers to the cited literature as indicated (Shi et al , FRBM 2021). We have moved the statement to the discussion section to avoid confusion.

Specific comments to the supplement

1.) In Fig. S2A the Western Blot is highly cut and re-assembled. The authors should show their results on one connected membrane instead.

(Now Fig. S3B).

Thank you for this suggestion. The result shown in original Fig.S2A was of course assembled from the same membrane (which can also be found in the submitted associated raw file). The reason why we cut and re-assembled the lanes was to display the cysteine mutants according to increasing amino acid number, as we originally loaded the samples in different order. We had actually also taken out a lane showing the C182S,C277S double mutant that we have now included as suggested by this reviewer in point 2 of the specific comments.

We have now included the whole blot without cutting and reassembly in the new Fig.S3B.

2.) The label of Fig. S3A appears to be wrong when looking at the expression and interaction profile of the p53 constructs.

We apologize for the confusion: the figure was indeed labeled incorrectly (the diamide-row was labeled + - + + + but should have been + + + - +). The labels have been corrected (now Fig. S4A).

3.) In Fig. S4C and S4D it is obvious that 1 out of 4 replicates from the WT-Dia group has a strong outlier. Was this samples included in the following analysis? If though, this might have strongly affected their conclusions drawn here. It the outlier is not included all diamide treated groups/p53 mutants appear quite similar in the PCA analysis (Fig. S4D).

(Now Fig. S5C and S5D)

We thank the reviewer for their comment, which made us look more carefully at the figure. We were also surprised to see one outlier out of the four replicates in the wt_Dia group shown in the PCA plot (now Fig. S5D), because the replicates seemed to be very reproducible as judged by Fig. S5A and S5B. It turns a minor mistake in the R scripts used for the PCA plot caused the outlier. As written in the Methods section, the raw MS data was filtered for reverse hits, standard contaminants and proteins for which less than two peptides were identified prior to further analysis. However, the PCA plot was made using the data before filtering for proteins with less than 2 peptides, which resulted in some noise compared to the other plots.

We have re-analyzed the data and made a new PCA plot using the filtered dataset (Fig.S5D), which indeed indicates a good reproducibility comparing replicates in each condition. We also that the dataset used is consistent for all plots and everything has been done properly.

4.) Although the authors shown in the lowest panel of Fig. S6 that 14-3-3 protein level could be reduced using siRNA, the upper panels, especially the left one, do not show any effect of si14-3-3-KD. Thus, the KD either did not work in the upper panels or has no effect.

(Now Fig.S7)

We realize that the 14-3-3-S-S-p53 band is quite faint in the non-reducing gels, and we have now indicated its position with an asterisk (*) in the left panels to aid the reader in the interpretation of the figure. The 14-3-3 band in the reducing gel (lower right panel) has now been indicated with a number symbol (#). We do not expect an effect of the 14-3-3 knock down in the upper right panel since it shows the reduced blot stained for Flag-p53. We hope that the effect of the 14-3-3 siRNA is clearer and that the figure now is easier to interpret.

Round 2

Reviewer 2 Report

The authors have carefully revised their manuscript to address my previous concerns and I would now recommend to accept their manuscript for publication.